# The Role of Reversible Phosphorylation of *Drosophila* Rhodopsin

**DOI:** 10.3390/ijms232314674

**Published:** 2022-11-24

**Authors:** Thomas K. Smylla, Krystina Wagner, Armin Huber

**Affiliations:** Institute of Biology, Department of Biochemistry, University of Hohenheim, 70599 Stuttgart, Germany

**Keywords:** GPCR signaling, rhodopsin phosphorylation, *Drosophila* eye, visual system, arrestin binding, receptor internalization, retinal degeneration

## Abstract

Vertebrate and fly rhodopsins are prototypical GPCRs that have served for a long time as model systems for understanding GPCR signaling. Although all rhodopsins seem to become phosphorylated at their C-terminal region following activation by light, the role of this phosphorylation is not uniform. Two major functions of rhodopsin phosphorylation have been described: (1) inactivation of the activated rhodopsin either directly or by facilitating binding of arrestins in order to shut down the visual signaling cascade and thus eventually enabling a high-temporal resolution of the visual system. (2) Facilitating endocytosis of activated receptors via arrestin binding that in turn recruits clathrin to the membrane for clathrin-mediated endocytosis. In vertebrate rhodopsins the shutdown of the signaling cascade may be the main function of rhodopsin phosphorylation, as phosphorylation alone already quenches transducin activation and, in addition, strongly enhances arrestin binding. In the *Drosophila* visual system rhodopsin phosphorylation is not needed for receptor inactivation. Its role here may rather lie in the recruitment of arrestin 1 and subsequent endocytosis of the activated receptor. In this review, we summarize investigations of fly rhodopsin phosphorylation spanning four decades and contextualize them with regard to the most recent insights from vertebrate phosphorylation barcode theory.

## 1. Introduction

Protein phosphorylation is a reversible molecular switch which its early study was awarded with the Nobel Prize in Physiology in 1992 for the groundbreaking work of Edmond H. Fisher and Edwin G. Krebs in the 1950s. Over 80 years of investigations found protein phosphorylation to be linked to the regulation of nearly every aspect of cellular life, for example, cell cycle control, gene regulation, enzyme activity, ion channel activity and receptor signaling. Among the numerous proteins regulated by reversible phosphorylation are G protein-coupled receptors (GPCRs) which are remarkable insofar as they represent targets for most of the drugs that have been approved for clinical use today [1]. Defective GPCR signaling has been shown to be involved in clinical conditions ranging from heart failure to psychiatric illnesses and retinal degeneration [2,3,4].

GPCRs are cell surface receptors which are activated by binding specific ligands, e.g., hormones, pheromones or neurotransmitters, resulting in intracellular signaling. Upon activation, GPCRs are typically phosphorylated at hydroxy groups of several serine and/or threonine residues in their intracellularly located C-terminal region by specific G-protein-coupled receptor kinases (GRKs) [5,6]. GPCR phosphorylation—in combination with glutamate and aspartate residues—generates a pattern of negative charges in the C-terminal domain of a GPCR that promotes binding of the inhibitory arrestin proteins [7,8,9,10]. In most cases, arrestin binding hinders interaction of the activated GPCR with its downstream G proteins and shuts off the signaling cascade. In order to return the receptor to its ground state, the phosphate groups are eventually removed which is achieved by specific phosphatases [11]. Besides termination of the signaling cascade by uncoupling the activated receptor from its G proteins, GPCR phosphorylation may contribute to the removal of activated receptors from the plasma membrane via endocytosis which leads to an attenuation of the signaling system by reducing receptor numbers at the cell surface and a turnover of receptor molecules. Although the principle of GPCR regulation by reversible phosphorylation presumably holds true for all GPCRs, the exact role of phosphorylation and its effect on receptor inactivation and internalization may differ somewhat among GPCRs.

In the present review we focus on the role of phosphorylation in fly rhodopsins and when appropriate compare it to vertebrate rhodopsins. Rhodopsins are prototypical GPCRs, and they play a major role in establishing the principles of GPCR signaling. They are abundant proteins in photoreceptor cells located at specialized cellular regions devoted to the detection of light, namely, the outer segments of vertebrate rod or cone photoreceptors and the rhabdomeres of photoreceptor cells in invertebrate compound eyes. Rhodopsins differ from other GPCRs because the activating “ligand” is a chemical compound that is permanently attached to the protein moiety, termed opsin. Opsins are seven transmembrane proteins that presumably evolved very early in one of the cnidarian–bilaterian ancestors [12]. Despite low-sequence similarities (22% amino acid identity between *Drosophila* Rh1 rhodopsin and bovine rhodopsin [13]), the overall structure of rhodopsin is surprisingly well-conserved between invertebrates and vertebrates (Figure 1A). Thus, data and hypotheses of rhodopsin’s molecular workings have traditionally been transferred from one model organism to another. The ligand is the 11-*cis*-retinal chromophore, or a derivative thereof, such as 11-*cis*-3-hydroxyretinal in flies, which is covalently bound via a Schiff’s base linkage to a lysine residue of the seventh transmembrane region. The chromophore lies approximately perpendicular to the seven α-helices of the opsin protein [14,15].

The inactive ground state of this GPCR is referred to as rhodopsin (R) while the activated state containing the chromophore in its all-*trans* configuration is called metarhodopsin (M). In vertebrate photoreceptors, activated metarhodopsin activates a heterotrimeric G protein termed transducin (Figure 1B). The α subunit of transducin in turn activates a phosphodiesterase (PDE) which results in the conversion of cyclic GMP (cGMP) to GMP. Cyclic nucleotide-gated ion channels in the plasma membrane of photoreceptor outer segments close in response to a drop in cGMP concentration, resulting in hyperpolarization of the photoreceptor membrane potential [16]. The visual signaling cascade in flies differs from this insofar as it utilizes a phosphoinositol pathway rather than cyclic nucleotide-mediated signaling (Figure 1C). Here, metarhodopsin activates a heterotrimeric G_q_ protein which activates phospholipase Cβ (PLCβ). Activated PLCβ hydrolyzes the membrane lipid phosphatidyl-inositol 4,5-bisphosphate (PIP_2_) resulting in the generation of diacylglycerol (DAG), inositol-trisphosphate (IP_3_) and a proton (H^+^). PLCβ activation also leads to changes in membrane tension, presumably inducing a reduction in the diameter of rhabdomeral microvilli and a measurable shortening of the entire rhabdomere [17]. These events trigger the opening of two transient receptor potential channels, TRP and TRPL, which leads to depolarization of the photoreceptor membrane potential [18,19].

In the following chapters we will first review some of the initial experiments on which todays understanding of rhodopsin phosphorylation is based. Subsequently, we will explain the impact that reversible phosphorylation has on the interaction of the arrestin–rhodopsin complex, complex internalization via endocytosis, and retinal degeneration. Finally, we summarize more recent findings on phosphorylation barcodes and discuss their potential implications.

## 2. History of Reversible Rhodopsin Phosphorylation in the Fly

### 2.1. Initial Studies of Light-Induced Rhodopsin Phosphorylation and Dephosphorylation

Following initial investigations of rhodopsin phosphorylation in vertebrates (frog, cattle, sheep) and higher invertebrates (octopus, squid), the first studies on rhodopsin phosphorylation in the photoreceptors of blowflies (*Calliphora vicina)* were published in 1984 [25]. Paulsen and Bentrop demonstrated that phosphorylation of rhodopsin is triggered by a photon absorption-induced state conversion from R to M, while reconversion from M to R correlated with dephosphorylation. Prior to this, another study postulated the existence of two additional inactive states—R_i_ and M_i_—as part of a model to explain the relationship between light absorption and photoreceptor cell depolarization in blowflies (Figure 2) [26]. Paulsen and Bentrop suggested that these two theoretically inactive states were represented by the two phosphorylated rhodopsin forms, R-P_n_ and M-P_n_ [25]. At this point, the data appeared to support investigations of bovine retinae, which implied that phosphorylation of activated M was part of the deactivation mechanism and resulted in an inability of M-P_n_ to further activate the transduction cascade [27]. However, this interpretation would later be disproven for invertebrates by experiments with unphosphorylatable fly rhodopsin which displayed an unaffected deactivation physiology and gave rise to one of several key differences between phototransduction in vertebrates and invertebrates [28]. Additionally, however, Paulsen and Bentrop’s findings suggested that photoregeneration of M-P_n_ to R-P_n_ via red light absorption in flies was not sufficient to completely recover from deactivation, implying that dephosphorylation had to occur to reconstitute activatable rhodopsin [25]. They estimated the kinetics of M phosphorylation and R-P_n_ dephosphorylation to have halftimes of 5 min and 20 s, respectively.

### 2.2. Phosphorylation Sites in Fly Rhodopsin

In a follow-up study from 1986, Bentrop and Paulsen reported the incorporation of 2.3 radioactive phosphates per molecule of rhodopsin on average after illumination with blue light with a calculated maximum of four phosphates per rhodopsin [29]. At this point, the opsin-coding gene *ninaE* from the fly *Drosophila melanogaster* had just been cloned independently by two groups and characterized regarding its homologies to vertebrate opsins [13,30]. Based on previous reports that sheep and cattle rhodopsins were phosphorylated mainly at seven potential phosphorylation sites within a small stretch close to the rhodopsin C-terminus, the groups of O’Tousa and Zuker both suggested six potential C-terminal phosphorylation sites present in *Drosophila* despite limited sequence conservation to vertebrate opsins in this region (Figure 3) [13,30,31,32,33]. A truncated version of opsin missing the last 18 amino acids including all six of these putative phosphorylation sites could indeed not be phosphorylated by rhodopsin kinase in vivo, confirming that the C-terminus harbored all sites for the light-induced phosphorylation of fly rhodopsin [28].

### 2.3. Identification and Characterization of Rhodopsin Kinase

In 1991, two *Drosophila* genes *gprk1* and *gprk2* were isolated by Cassill and colleagues in an attempt to identify putative G protein-coupled receptor kinases (GPRKs) that may be involved in the regulation of GPCR-mediated cellular processes [35]. These genes were found to possess high sequence similarity to the β-adrenergic receptor kinase from vertebrates of around 60 to 80%, including a highly conserved kinase domain but were originally not discussed as putative rhodopsin kinases. The first biochemical studies regarding rhodopsin kinase in fly photoreceptors came from *Musca domestica* in 1992 [36]. Doza and colleagues used a partially purified fraction of eye extract to characterize its properties regarding the ability to phosphorylate rhodopsin and reported stringent dependence on R to M conversion for rhodopsin binding and enzymatic activity of the proposed meta-rhodopsin kinase. Interestingly, investigations from *Calliphora* implied that activity and interaction of rhodopsin kinase with metarhodopsin were promoted by binding of the cytoplasmic protein arrestin 2 to activated rhodopsin [37]. This notion was later connected with findings in vertebrates under the assumption that arrestin binding might lead to conformational rearrangements in rhodopsin, unmasking specific regions (e.g., binding sites, phosphorylation sites) [38]. Over ten years after its identification, Lee and Montell reported that GPRK1 physically interacted with rhodopsin and was indeed rhodopsin kinase [39]. This study also found GPRK1 to harbor an additional C-terminal pleckstrin homology (PH) domain known to bind to phosphoinositides.

### 2.4. Identification and Characterization of Rhodopsin Phosphatase

The *rdgC* gene of *Drosophila* had been predicted to encode a phosphatase with multiple Ca^2+^-binding EF hand motifs in the early 1990s [40]. Soon thereafter, it was shown to act in vitro on phosphorylated rhodopsin and metarhodopsin as a substrate in a Ca^2+^-dependent manner by Byk and colleagues [41]. The dephosphorylation of M-P_n_ was shown to be inhibited in the presence of arrestin, suggesting that arrestin’s interaction with metarhodopsin serves as a protection mechanism against becoming dephosphorylated by RDGC under physiological conditions. Plangger and colleagues showed that rhodopsin phosphatase existed physiologically at least partially in a membrane-associated form and confirmed that it did not discriminate between R-P_n_ and M-P_n_, but instead was prevented from using M-P_n_ as a substrate by its interaction with arrestin 2 [42]. In vivo evidence of RDGC as a physiologically significant rhodopsin phosphatase was later presented by way of experiments that used consecutive illuminations with blue and orange light to induce rhodopsin phosphorylation and subsequent dephosphorylation in *Drosophila* [28]. As was expected from initial studies in *Calliphora* [25], photoregeneration from M-P_n_ to R-P_n_ triggered a significant reduction in the portion of phosphorylated rhodopsin in wild type flies, implying phosphatase activity. However, in *rdgC* mutants the amount of incorporated radioactive phosphate was even higher in blue- and subsequently orange-illuminated flies than in wild type flies immediately after blue illumination, suggesting that RDGC was indeed a rhodopsin phosphatase and that its absence resulted in hyperphosphorylated rhodopsin [28]. In 2001, RDGC was further characterized and was reported to possess an N-terminal IQ-motif which interacted directly with the Ca^2+^-sensing calmodulin, an interaction that was enhanced in the presence of Ca^2+^ ions and strictly required for phosphatase activity in vivo [43]. Lee and Montell also described three RDGC isoforms of distinct molecular weights, of which the two shorter ones appeared to be eye-specific. These isoforms were later traced back to a combination of varying transcription start sites, alternative splicing and start codon usage, resulting in variations of the N-terminal region that affect the subcellular localization of the different RDGC isoforms [44]. Strauch and colleagues found the two longer isoforms to be fatty acylated at the N-terminus, leading to their membrane association, whereas the shortest isoform was mainly cytoplasmic. Additionally, these three isoforms were demonstrated to serve a redundant phosphatase function regarding the dephosphorylation of rhodopsin albeit with varying physiological contributions [45]. Aside from rhodopsin, RDGC also dephosphorylates the Ca^2+^-dependent photoreceptor ion channel TRP at a serine residue at position 936 which results in a form of light adaptation that allows the processing of stimuli with a high-temporal resolution in bright conditions [46]. RDGC is classified as a protein phosphatase with EF hands (PPEF) of which there are also vertebrate and human orthologues [47]. However, while they are expressed in various neuronal tissues, including photoreceptor cells, these PPEFs have not been found to be implicated in rhodopsin dephosphorylation [48,49,50].

### 2.5. Rhodopsin Regeneration Cycle in Photoreceptors

The first comprehensive theory with regard to a concerted regeneration cycle involving rhodopsin phosphorylation, interaction with arrestins and dephosphorylation that would prove to be most consistent with previous and future findings in flies arose in the early 1990s [41]. Therein, Byk and colleagues proposed a regulatory cycle in which blue light absorption triggered the conversion of R to M which is subsequently phosphorylated to M-P_n_. Binding of arrestins to M-P_n_ results in deactivation and inhibits M-P_n_ dephosphorylation by RDGC until M-P_n_ is reconverted to R-P_n_ by photoregeneration [41]. The concomitant release of arrestins from R-P_n_ enables RDGC to dephosphorylate R-P_n_ to R which closes the cycle. In light of later studies, the order of arrestin binding and rhodopsin phosphorylation has been adjusted accordingly in the current model (Figure 4). In detail, Plangger and colleagues reported kinetics for both the phosphorylation of metarhodopsin and its binding to arrestin 2: While metarhodopsin’s phosphorylation was recorded with a half-life of 9 min in vitro—corroborating previous results [25]–arrestin 2 binding could not even be resolved temporally in their setup [42]. This suggested arrestin 2 binding occurred seconds after the conversion from R to M and, thus, long before metarhodopsin was phosphorylated to a substantial degree. Similar time frames have also been reported for *Drosophila*, where arrestin 2 binding-dependent deactivation was measured with time constants between 100 ms and seconds [51,52,53,54,55]. Interestingly, *Drosophila* arrestins become phosphorylated themselves shortly after illumination by a Ca^2+^/calmodulin-dependent kinase (CamKII) [56,57,58]. Initial investigations proposed that the phosphorylation of arrestin 2 is implicated in the release of arrestin 2 after reconversion of the visual pigment from M-P_n_ to R-P_n_ [59,60]. However, following studies concluded that phosphorylation of arrestin 2 had no significant effect on its association with metarhodopsin or its dissociation from rhodopsin [61]. In their study from 1993, Byk and colleagues additionally discussed the connection between arrestin binding and the prolonged depolarization afterpotential (PDA) [41]. The PDA is a phenomenon observed in electroretinogram recordings when white-eyed flies are illuminated with blue light. In this case, the depolarization of photoreceptor cells does not return to baseline when the light is switched off and the cells remain depolarized until the eyes are illuminated with orange or red light [62,63]. As we understand it today, blue light of about 455 nm converts ca. 70% of the visual pigment from R to M which represents the maximum conversion ratio according to the absorption properties of *Drosophila*’s main rhodopsin (Rh1) (Figure 4) [64,65]. Due to its around five times lower abundance compared to rhodopsin, the arrestins present in photoreceptor cells can only inactivate around 30% of M generated under these conditions and the remaining active M keeps triggering the signaling cascade, resulting in receptor depolarization [66,67]. Orange or red light of 590 nm or higher converts 98–100% of M back to inactive R and hence terminates the PDA [64,67]. A limiting concentration of free arrestin for M inactivation turned out to be a much more elegant explanation for the phenomenon of the PDA than earlier explanations, including limited availability of local ATP or activity and access of rhodopsin kinase as a cause for limited rhodopsin phosphorylation [25,41]. This hypothesis was substantiated by experiments showing that flies with reduced concentrations of arrestin molecules (e.g., arrestin mutants) enter a PDA much faster than wild type flies and that rhodopsin phosphorylation alone is insufficient to induce deactivation [53,64].

## 3. The Role of Rhodopsin Phosphorylation in the Fly

### 3.1. The Role of Rhodopsin Phosphorylation for Arrestin Binding

Over the past decades, arrestin proteins have been studied alongside GPCRs, since they are equally well conserved across evolutionarily separate species [12]. Similar to vertebrates, invertebrate arrestins mediate deactivation of G protein signaling and GPCR internalization which is recurrently referred to as “canonical” arrestin function and has been elucidated by numerous studies especially on rhodopsin and beta-adrenergic receptors (β2AR). In addition, specifically β-arrestins in vertebrates have been described to act as scaffolds or transducers for diverse signaling pathways often summarized as “non-canonical” arrestin signaling. β-arrestins have been found to control cellular processes, such as cell migration, apoptosis, chemotaxis and differentiation, for example, by means of MAPK/ERK/JNK signaling, non-receptor tyrosine kinase (nRTK) signaling, or Wnt signaling [73,74,75,76]. Interestingly, β-arrestins have further been reported to prolong G protein-mediated signaling instead of promoting its deactivation [77]. Additionally, so-called arrestin-coupled receptors (ACRs) are defined as a subcategory of seven transmembrane receptors (7TMR) that do not signal through G proteins as opposed to prototypical GPCRs [78]. Instead ACRs (i.e., D6R and C5aR2) use β-arrestins as transducers for ERK signaling and neutrophil mobilization. Of note, these two arrestin–ACR complexes are independent of ACR C-terminus phosphorylation [78]. Furthermore, there is evidence that arrestins in complex with D6R or C5aR2 take on distinct conformations. In *Drosophila*, non-visual β-arrestin (encoded by *krz*) was found to be involved in the regulation of different signaling pathways, such as Notch, MAPK/Toll, and Hedgehog [79,80,81]. While these arrestin-dependent signaling pathways can be assigned to the “non-canonical” arrestin functions, for the visual arrestins in *Drosophila* no “non-canonical” functions have been described thus far.

Flies express two photoreceptor-specific arrestins, similarly to the two known splice variants of visual S-arrestin proteins in vertebrate rod cells [82]. Originally known as phosrestin I and phosrestin II in *Drosophila*, they are now termed arrestin 2 and arrestin 1, respectively [57,58,83,84]. As is the case for the proteins generated from the two splice variants of S-arrestin in vertebrates, the main difference between these two *Drosophila* proteins lies in a shortened C-terminus present in arrestin 1 (39 kDa) compared to arrestin 2 (49 kDa) [58,83]. Arrestin 2 is about five times more abundant than arrestin 1 and plays a major role in the deactivation of the visual signaling cascade [53,58,85]. Accordingly, *arr2* loss-of-function mutants display severe defects in receptor deactivation. While mutations of *arr1* appear to have no significant effect on receptor deactivation on their own, the defect of *arr2* mutants is enhanced in *arr1*; *arr2* double mutants, highlighting an overlapping function [53].

For vertebrate rod outer segment (ROS) disk membranes, it has been reported early on and repeatedly that the rhodopsin-binding arrestins specifically bind to the phosphorylated M state [6,86,87,88,89]. In the 1990s, it became increasingly clear that—unlike in vertebrates—phosphorylation of M by rhodopsin kinase was not necessary for the deactivation of the signaling cascade in fly photoreceptors. Thus, why do flies phosphorylate rhodopsin in the first place? In flies, several findings have been published, contradicting the notion that metarhodopsin phosphorylation is required for initial arrestin binding. As mentioned above, the most obvious argument is a temporal discrepancy in that arrestin 2 binding occurs on a scale of (milli-)seconds whereas rhodopsin phosphorylation takes several minutes [25,42,51,52,53,54,55]. The first to mention the physical binding of arrestin 2 (originally known as the 48K protein) to rhodopsin were Bentrop and Paulsen in 1986 [29]. They observed a light-dependent soluble phosphorylated 48 kDa protein that became reversibly membrane-bound in vitro only in the presence of rhodopsin in its M state. However, they found that binding of arrestin 2 was independent of ATP and, thus, independent of phosphorylation [29]. In further investigations, Vinós and colleagues reported that deletion of the entire C-terminal region resulted in phosphorylation-deficient metarhodopsin as expected [28]. Strikingly, arrestin 2-mediated deactivation kinetics were not affected by this truncation at all. Still, this study also discussed the idea that the rhodopsin C-terminus might act as an autoinhibitory domain for arrestin binding which vacates the binding site only as a consequence of its phosphorylation [28]. Yet, even single point mutations of the C-terminal serine and threonine residues had no apparent effect on the light-induced association and dissociation behavior between arrestin 2 and rhodopsin [61,90]. A later study found that increasing or decreasing the rhodopsin phosphorylation state by overexpression of a functional or dominant negative variant of GPRK1, respectively, appeared to have no correlative effect on arrestin 2 binding [39]. Since both cases resulted in a slight loss in bound arrestin 2, one could claim that there may be an optimal phosphorylation state that—when disturbed—affects arrestin 2 binding negatively. However, Lee and Montell favored the interpretation that GPRK1 might have a phosphorylation-independent role affecting arrestin 2–rhodopsin binding [39]. With regard to complex dissociation in response to orange illumination, there were no apparent differences. Interestingly, this study observed an effect on the amplitude of the photoresponse in which higher GPRK1 activity—and thus higher rhodopsin phosphorylation levels—correlated with lower amplitudes and vice versa [39].

Nevertheless, there have also been studies that explicitly corroborate a particular role of rhodopsin phosphorylation for its interaction with arrestins. In one study regarding retinal degeneration, Alloway and colleagues reported the necessity of the phosphorylatable rhodopsin C-terminus for the persistence of so-called stable arrestin–rhodopsin complexes, inferring that an interaction between a C-terminally truncated rhodopsin and arrestin 2 was possible but transient [91]. Interestingly, stable complexes were also involved in a significantly reduced deactivation rate in *rdgC* mutants which was attributed to the phosphorylation state of rhodopsin by truncation of its C-terminus [28]. This implies that the phosphorylation state of metarhodopsin in *rdgC* mutants may enhance arrestin binding and/or hinder arrestin release from rhodopsin, slowing deactivation down. Moreover, *rdgC* mutants enter a PDA with much lower levels of illumination than wild type, similar to *arr2* mutants [28]. This finding may be explained by the assumption that arrestin, which is tightly bound to rhodopsin, is no longer available for quenching activated M, i.e., the amount of effectively free arrestin is significantly reduced. By further investigation of C-terminal deletion and single point mutants in rhodopsin, another study reported a minor reduction in binding efficacy upon blue illumination for arrestin 2 but a significant effect for arrestin 1 [92]. This difference in the role of rhodopsin phosphorylation for the function of the two arrestins was also supported by results investigating the subcellular translocation of arrestins from Shieh and colleagues, who used GFP fusion proteins of both arrestin 1 (Arr1:eGFP) and arrestin 2 (Arr2:eGFP) to measure changes in rhabdomeric fluorescence in correlation to rhodopsin activation and its ability to become phosphorylated [93]. The study found that the translocation of Arr2:eGFP to rhabdomeres upon illumination was not affected by C-terminal truncation of rhodopsin or mutation of all six putative phosphorylation sites. In contrast, Arr1:eGFP translocated to the rhabdomeres only in the presence of wild type but not phosphorylation-deficient metarhodopsin, where it remained mostly in the cytoplasm [93]. Given the differences in sequence between arrestin 1 and arrestin 2, it seems possible that rhodopsin phosphorylation affects the binding to each of these proteins differently. To date, the controversy regarding rhodopsin phosphorylation and its role in arrestin binding has not been completely resolved. While it may play a role in particular for the binding of arrestin 1, it seems that rhodopsin phosphorylation is not needed for arrestin 2-mediated deactivation of M in flies, contrary to what has been reported for vertebrate rhodopsin or other GPCRs.

### 3.2. The Role of Rhodopsin Phosphorylation for Receptor Internalization

In the absence of the Ca^2+^-dependent rhodopsin phosphatase RDGC, rhodopsin has been shown to be excessively phosphorylated by means of massive incorporation of radioactive ^32^P [28]. Based on a highly similar phenotype (which will be described in detail later) it has also been argued that mutants with a lack of Ca^2+^ influx—specifically *norpA* (encoding PLCβ)—display the same excess of rhodopsin phosphorylation [61,94]. This hyperphosphorylation of rhodopsin’s C-terminus can be monitored by the inability of a monoclonal antibody to target the C-terminal epitope of rhodopsin (DSHB, 4C5), as evidenced by a total lack of signal in immunoblots of *rdgC* mutant flies after white illumination [45,95]. This phenomenon had originally been described in retinal tissue sections of blue-illuminated *norpA* mutants as “epitope masking” of the rhodopsin C-terminus by arrestin 2 [96]. However, arrestin 2 binding appears to be an insufficient explanation, considering the widely accepted molecular ratio of arrestin 2 to rhodopsin of about 1:5—as calculated by the R to M conversion needed to induce a PDA in *Calliphora* and *Musca* [66,67]. In light of these numbers, around 80% of rhodopsin molecules in photoreceptor cells would remain unmasked at all times. Instead, excessive phosphorylation of the epitope seems a more plausible interpretation for its masking.

In *rdgC* and *norpA* mutants, hyperphosphorylation has been suggested to induce the formation of arrestin 2–rhodopsin complexes that are more stable than their physiologically phosphorylated counterparts and undergo massive endocytic internalization [90,91]. These internalized complexes ultimately accumulate in late-endosomes as a consequence of saturation of the lysosomal system [97]. Further evidence regarding the connection of rhodopsin phosphorylation, arrestin binding and internalization came from studies investigating light-induced photoreceptor degeneration [98]. Lee and Montell observed a gradual decrease in rhodopsin protein levels in flies exposed to continuous illumination, while the levels of several other signaling proteins (G_q_, PLCβ, TRP, Arr2, INAD, PKC) remained unchanged [98]. This decline in rhodopsin correlated with the degeneration phenotype monitored by a steady decrease in photoresponse. Lee and Montell reasoned that continued degradation of rhodopsin was causing the visual impairment and found significantly ameliorated phenotypes in flies expressing C-terminally truncated rhodopsin or *arr2* mutants [98]. Taken together, these findings portray a clear role of phosphorylation in rhodopsin’s C-terminus for stable interactions with arrestin 2, their joint internalization, and subsequent degradation of rhodopsin under unphysiological conditions, such as *rdgC* and *norpA* mutations or continuous illumination.

To understand the significance of rhodopsin phosphorylation in a physiological context, we have to consult additional studies: Rhodopsin internalization as part of rhabdomere turnover in *Drosophila* photoreceptors had been proposed by Stark and colleagues as early as 1988 [99]. In their study, they argued that there was a “diurnal rhythm of visual pigment” that could be measured by micro-spectrophotometry and electron microscopy. The absence of multivesicular bodies (MVBs), specifically in the outer photoreceptor cells of a nonsense mutant of *ninaE* (known as *outer rhabdomeres absent* (*ora*)), suggested that rhodopsin was the main cargo of internalized membranes that later formed MVBs [99]. As shown for vertebrates before, arrestin 2 physically interacts with clathrin in vitro as well as with the endocytic adaptor complex AP-2 [90,100]. The hypothesis of clathrin-mediated endocytosis of rhodopsin was corroborated by multiple studies in which a decrease in cytoplasmic rhodopsin was documented in flies suffering from a loss of dynamin function (*shibire*) [90,92,101]. In a genetic screen, the eye-enriched tetraspanin-encoding gene *sunglasses* (*sun*) was shown to be involved in the light-dependent lysosomal degradation of rhodopsin [102]. Interestingly, Xu and colleagues claimed that the illumination-induced reduction in rhodopsin protein was not dependent on the presence of arrestin 2 and, thus, had to rely on another mechanism. By means of recording time courses of the subcellular localizations of arrestins 1 and 2 in pupal photoreceptor cells, Satoh and Ready found that endocytosed rhodopsin was mainly co-localizing with arrestin 1 in response to light [92]. Moreover, they found that the existence of rhodopsin-containing MVBs was strictly dependent on arrestin 1, not arrestin 2. Their experiments using phosphorylation-deficient rhodopsin mutants indicated that physiological rhodopsin endocytosis requires interaction between arrestin 1 and metarhodopsin which in turn depends on phosphorylation of rhodopsin’s C-terminus [92]. Satoh and Ready concluded that while the two fly arrestins have overlapping functionality, in a physiological context arrestin 1 specializes in phosphorylation-dependent rhodopsin endocytosis, whereas arrestin 2 is mainly responsible for phosphorylation-independent quenching of the phototransduction cascade. Since arrestin 1 lacks C-terminal motifs for both clathrin and AP-2, the precise endocytic mechanism remains to be determined [92]. Of interest, Shieh and colleagues did not observe vesicular localization of internalized Arr1:eGFP in pupal photoreceptors in adults, arguing that light-induced endocytosis of arrestin 1–metarhodopsin complexes may be a process relevant mainly during photoreceptor development [93]. In summary, several studies have shown that rhodopsin internalization is both an integral part of physiological rhodopsin turnover as well as a hallmark of certain mutant phenotypes. Depending on the context, either arrestin 1 or arrestin 2 binding induces endocytosis of rhodopsin-containing membranes. However, both interactions appear to be influenced by the (hyper-)phosphorylation status of rhodopsin’s C-terminus.

### 3.3. The Role of Rhodopsin Phosphorylation for Photoreceptor Degeneration

Flies that are null mutants in the *rdgC* (*retinal degeneration C*) gene have originally been described to suffer from age- and light-dependent retinal degeneration, resulting in a severely defective ultrastructure after 8 days in a 12-h light/dark cycle [103]. While this degenerative phenotype required photoactivation of R to M, it was independent of subsequent phototransduction steps, as neither loss of the heterotrimeric G protein α subunit (*dgq* mutant) nor loss of PLCβ activity (*norpA* mutant) was able to prevent degeneration in *rdgC* mutants [28,103]. Similar to *rdgC* mutants, flies null mutants in *norpA* also suffered from light-dependent photoreceptor degeneration leading to severe structural defects within 6 to 10 days of a 12-h light/dark cycle [61,104]. For *rdgC* degeneration, Vinós and colleagues demonstrated a strong suppression by truncation of rhodopsin’s phosphorylation-site-harboring C-terminus, suggesting that hyperphosphorylation of rhodopsin—due to a lack of phosphatase activity—played a key role in this degeneration pathway [28]. This interpretation was later corroborated by Lee and Montell who reported photoreceptor degeneration in the presence of a calmodulin interaction-deficient transgenic RDGC variant which was unable to sense light-induced Ca^2+^ influx and, thus showed no phosphatase activity in response to illumination [43]. The same conclusion was reached for *norpA* mutants, in which retinal degeneration was also significantly ameliorated by removing rhodopsin’s C-terminus [91,105]. Two parallel studies investigating the degenerative phenotypes in *rdgC* and *norpA* mutants concluded that both are the consequence of massive internalization of stable arrestin 2–rhodopsin complexes [90,91]. While Kiselev and colleagues could rescue photoreceptor degeneration in *rdgC* mutants with an *arrestin 2* loss-of-function allele, Alloway and colleagues demonstrated that degeneration in *norpA* mutants was prevented by removing arrestin 2 or reducing the amount of rhodopsin by rearing flies on vitamin A-deprived food. Additionally, degeneration in *norpA* mutants was significantly slowed by changing the light quality of the 12 h light/dark cycle from white to orange, shifting the pigment ratio continuously away from the M state towards the R state [106]. Overall, Kiselev and colleagues argue that *rdgC* mutants lack phosphatase activity to prevent stable arrestin–rhodopsin complexes directly [90]. In contrast, the interpretation regarding *norpA* mutants is more elaborate: Apart from RDGC activity, *norpA* mutants also lack CaMKII activity due to their inability to open TRP Ca^2+^ channels. This results in unphosphorylated arrestin 2 which interacts strongly with clathrin, leading to increased endocytosis [90]. Still, Kristaponyte reported that loss of CaMKII activity may only enhance, not initiate the degenerative phenotype, as was shown in experiments with an inhibitory peptide [61].

Taken together, these results imply that M-P_n_ is clathrin-dependently internalized as a result of its interaction with arrestin 2 and that hyperphosphorylation in the *rdgC* and *norpA* mutants is etiological for retinal degeneration due to long-lived arrestin–rhodopsin complexes. Under physiological conditions, the role of arrestins to facilitate rhodopsin internalization allows for a turnover of specifically the activated metarhodopsin, to which arrestin 1 and 2 bind. Conversely, in mutants or unphysiological conditions (e.g., blue-light illumination) in which a large amount of arrestin is bound to R or M this mechanism has detrimental effects, resulting in massive rhodopsin internalization that presumably overloads the endolysosomal system and leads to photoreceptor cell degeneration.

## 4. The Role of Phosphorylation in Arrestin–GPCR Complexes

### 4.1. Molecular Conformation of Activated Arrestin

The interaction of arrestins with GPCRs is accompanied by very specific conformational changes which have been elucidated in detail by several independent groups. Arrestin proteins consist of a bi-lobed structure connected by a quasi-hinge and a C-terminal tail that keeps the arrestin in its inactive conformation [107]. Upon binding of an arrestin to an activated GPCR, the negative charges of the phosphorylated receptor C-terminus displace the arrestin’s own C-tail, thus initiating a transition into the arrestin’s activated conformation. This entails a relative rotation of the two lobes by about 20°, a destabilization of the central core, opening of a cleft for the accommodation of the GPCR’s second intracellular loop (ICL2) helix, and the adoption of the arrestin’s negatively charged finger loop into an opening of the GPCR’s transmembrane bundle [23,108,109,110,111,112]. This structural information has been gathered in vertebrates by pseudo-activation of arrestins through C-tail truncation, activation of arrestins with a phosphopeptide of the vasopressin receptor (V2R), as well as crystallization of an arrestin–rhodopsin complex. Beyond this general structural model, studies with a β2AR–V2R chimera describe a (transient) state of the arrestin–GPCR complex in which the arrestin interacts with only the C-tail of the GPCR [113]. Additionally, in recent reports an arrestin was found to bind to the neurotensin receptor (NTSR1) in a conformation that differs significantly from that of the arrestin–rhodopsin complex [114]. Here, the arrestin rotated clockwise by about 90° on an axis perpendicular to the membrane plane when looking at it from the extracellular space. Taken together, arrestin–GPCR formation is far from being completely understood. The range of possible conformations, however, suggest that each variation might be linked to very specific downstream signaling [115]. In contrast to vertebrates, invertebrate arrestins also bind to unphosphorylated metarhodopsin. Studies with squid arrestin–rhodopsin complexes suggest that the C-tail of invertebrate arrestin does not lock arrestin in its inactivated conformation and, thus, does not need to be released by the interaction with the negatively charged rhodopsin C-terminus [116]. However, since squid arrestin has a remarkably long C-tail with little resemblance to other species, this does not have to be true for all invertebrates. Structures of *Drosophila* arrestin–rhodopsin complexes have not been resolved yet.

### 4.2. Phosphorylation Barcodes

Vertebrate GPCRs—including rhodopsins—are phosphorylated C-terminally with certain patterns that are then recognized and bound by distinct regions on arrestins [7,117]. These positively charged “pockets” of arrestins have recently been described in more detail based on structural information from an arrestin–rhodopsin complex and are highly conserved between vertebrates and invertebrates (Figure 5A) [8]. The corresponding phosphorylation barcodes on the C-terminus of rhodopsin or the vasopressin receptor V2R have been described to take the form of the motif Px(x)PxxP/E/D, where “P” represents a phosphorylated serine or threonine, “x” any amino acid except proline and “E” or “D” the one letter abbreviations of glutamate or aspartate, respectively. Such a motif could also potentially be generated within the C-terminus of fly rhodopsin, considering that it appears to be phosphorylated on at least two positions on average after light-induced activation (Figure 5B) [29]. Further studies on vertebrate GPCRs revealed that different phosphorylation patterns in the C-terminal region favor a wide variety of arrestin conformations, suggesting the effects of these phosphorylation barcodes on signaling of internalized arrestin-GPCR complexes [10]. Interestingly, two of the three positively charged amino acids (R19 and R172 according to the mural S-arrestin sequence) supposedly stabilizing the first negative charge of the proposed GPCR Px(x)PxxP/E/D motif in vertebrates are absent in *Drosophila* arrestins 1 and 2 (Figure 5A). Recently, Mayer and colleagues described a much wider rhodopsin phosphorylation motif, in which they assigned distinct functions to the different regions within the receptor’s C-terminus, such as key sites, modulator sites and inhibitory sites [118]. However, this detailed model is not only less restrictive but at the same time more species-specific, since it is highly applicable for other vertebrates and their various classes of GPCRs, but it not as easily translated to fly rhodopsin due to low-sequence similarities (Figure 5C).

In light of the findings regarding GPCR phosphorylation barcodes, it may be speculated that hyperphosphorylation of *Drosophila* rhodopsin, as observed, for example, in *norpA* or *rdgC* mutants, facilitates the binding of arrestin in an unusual conformation with high-affinity. This, in turn, may affect endocytosis of arrestin–rhodopsin complexes or downstream events, such as the lysosomal degradation of rhodopsin. Indeed, Satoh and Ready suggested that the hyperphosphorylation of rhodopsin affects processes, such as clathrin-mediated endocytosis, by the formation of stable arrestin 2–rhodopsin complexes which under physiological phosphorylation conditions are much more transient [92]. Additionally, Shieh and colleagues discussed the possibility of differentially phosphorylated metarhodopsin in the context of different trafficking patterns of arrestin 1–rhodopsin complexes which they observed between pupal and adult photoreceptors [93]. Ever increasing structural information about complexes formed by GPCRs with their corresponding arrestins may shed light on the underlying mechanisms of these varying pathways.

## 5. Concluding Remarks

In vertebrate photoreceptors, rhodopsin phosphorylation per se facilitates the inactivation of the activated receptor by enhancing arrestin binding. In *Drosophila*, rhodopsin phosphorylation does not seem to affect receptor inactivation but plays an important role for endocytic rhodopsin internalization. In the fly, rhodopsin internalization is required for the turnover of the visual pigment that may help to refresh worn out rhodopsin molecules. In vertebrate rod photoreceptors, this is achieved by transporting “old” rhodopsin to the tip of the rod and then removing the tip by phagocytosis via adjacent pigment epithelium [119]. Interestingly, in fly photoreceptors a large proportion of internalized rhodopsin is degraded in lysosome and replenished by newly synthesized protein, but some of the internalized rhodopsin is not degraded and is instead recycled back to the photoreceptor membrane [120]. This recycling requires the retromer complex and its failure results in photoreceptor degeneration. How can the retromer complex distinguish between recycling and lysosome-bound rhodopsin? Possibly, the phosphorylation state of rhodopsin may contribute to this distinction either by way of certain arrestin conformations or directly—as suggested for other GPCRs and their interaction with the SNX27–VPS26–retromer complex [121,122]. In this context, it is of note that the rhodopsin phosphatase RDGC is present not only in its membrane-bound form to potentially dephosphorylate rhodopsin in the photoreceptor membrane, but also as a splice variant that encodes a soluble RDGC variant [44].

Several mutants, for example *rdgC* and *norpA*, affect the phosphorylation of rhodopsin and show severe retinal degeneration. When the rhodopsin phosphatase RDGC is absent or cannot be activated by Ca^2+^, as is the case in the *norpA* mutant, hyperphosphorylation of rhodopsin is observed. There is convincing evidence that this hyperphosphorylation results in enhanced endocytosis of rhodopsin via high-affinity arrestin binding and constitutes the prime reason for light-induced death of photoreceptor cells. In the simplest hypothesis, the sheer amount of a highly abundant protein such as rhodopsin in the endolysosomal system may overload this system, ultimately leading to degeneration. Alternatively, arrestin–rhodopsin complexes that may not dissociate when arrestin is bound to hyperphosphorylated rhodopsin could actively trigger apoptosis. Whether or not changes in the phosphorylation pattern can have similar negative effects in vertebrate rhodopsins or hormone receptors is not yet known, but evidence is accumulating that changes in the phosphorylation pattern of hormone receptors may well affect downstream signaling [10,117,118,123].

## Figures and Tables

**Figure 1 ijms-23-14674-f001:**
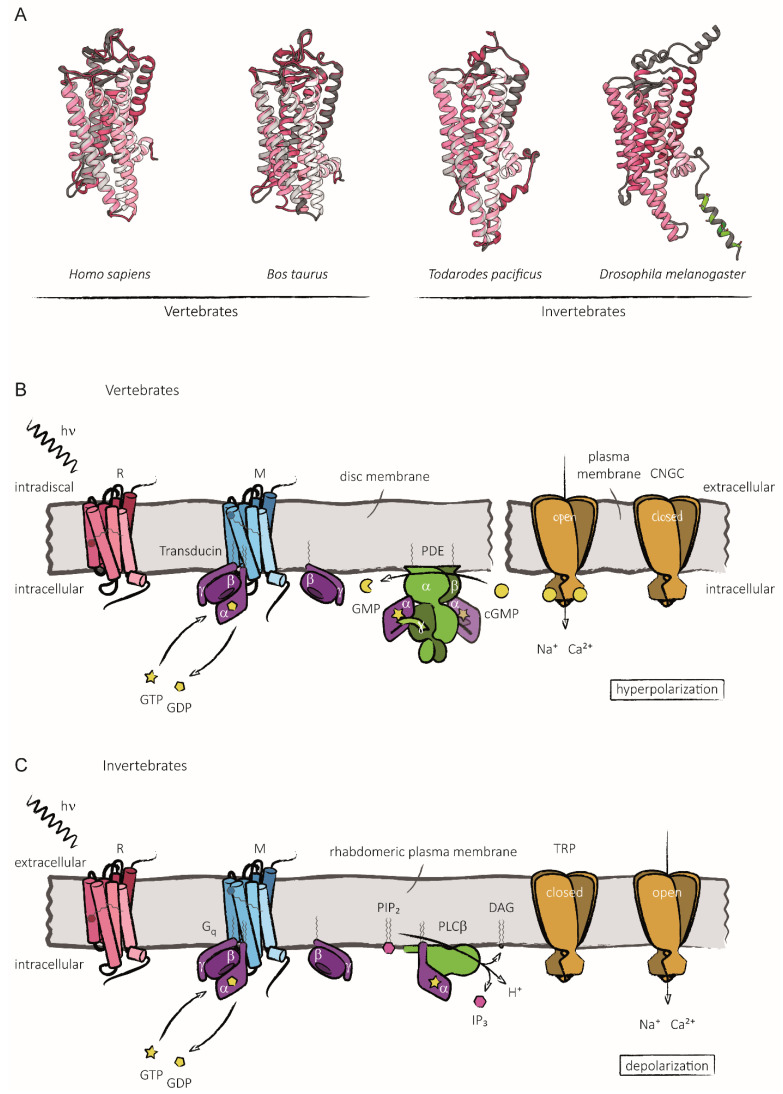
Schematic representation of phototransduction cascades in vertebrates and invertebrates. (**A**) Rhodopsin proteins as ribbon models in shades of fuchsia from *Homo sapiens* (human), *Bos taurus* (cattle), *Todarodes pacificus* (squid) and *Drosophila melanogaster* (fruit fly) based on structural predictions from AlphaFold DB [20]. Predicted structures of squid, cattle and human are aligned with published X-ray crystal structures 2Z73 [21], 3C9L [22] and 4ZWJ [23] from RCSB PDB [24], respectively, displayed in shades of gray to visualize AlphaFold’s accuracy. Protein crystal structures commonly omit rhodopsin’s flexible C-terminus which has thus been cropped from the corresponding models predicted by AlphaFold as well. The C-terminus of AlphaFold’s *Drosophila* prediction is included despite low confidence to illustrate phosphorylatable serine (light green) and threonine (dark green) residues as a ball and stick representation. (**B**) Exemplarily depicted is the phototransduction cascade in the rod cells of humans. In the disc membrane, photon absorption results in the transformation of opsin-bound 11-*cis*-retinal to all-*trans*-retinal and subsequently the conversion from rhodopsin (R) (fuchsia) to metarhodopsin (M) (blue). M interacts with the heterotrimeric G protein (transducin) (purple), promoting the exchange of GDP (yellow pentagon) for GTP (yellow star) within the α subunit (Gα). Gα-GTP is separated from the βγ subunits of transducin and binds to phosphodiesterase (PDE) (green) whereupon it partially retracts the catalysis-inhibiting γ subunit of PDE. Thusly activated by Gα-GTP, PDE catalyzes the hydrolysis of cGMP (yellow circle) to GMP (yellow major sector), reducing the local cGMP concentration. The decrease in cGMP levels leads to its disengagement from cyclic nucleotide-gated channels (CNGCs) in the plasma membrane (orange), resulting in channel closing. Without further Na^+^ and Ca^2+^ ion influx via open CGNCs, the vertebrate rod cell hyperpolarizes. (**C**) The phototransduction cascade of flies serves as an example of invertebrates. In rhabdomeric plasma membranes, photon absorption transforms opsin-bound 11-*cis*-3-hydroxyretinal to all-*trans*-3-hydroxyretinal whereupon R (fuchsia) is converted to M (blue). The interaction of the heterotrimeric G protein (Gq) (purple) with M leads to the exchange of GDP (yellow pentagon) for GTP (yellow star) and the release of Gα-GTP. As a result of its binding, Gα-GTP activates the catalytic activity of PLCβ which in turn cleaves phosphatidylinositol-4,5-bisphosphate (PIP_2_) (magenta) into diacylglycerol (DAG) and inositol-3-phoshate (IP_3_) (magenta hexagon) accompanied by the release of one proton (H^+^). This results in the opening of membrane-embedded transient receptor potential (TRP) channels (orange) and concomitant Na^+^ Ca^2+^ influx which ultimately depolarizes the invertebrate photoreceptor cell. Grey zigzag lines represent hydrocarbon-based chains, e.g., lipid modifications.

**Figure 2 ijms-23-14674-f002:**
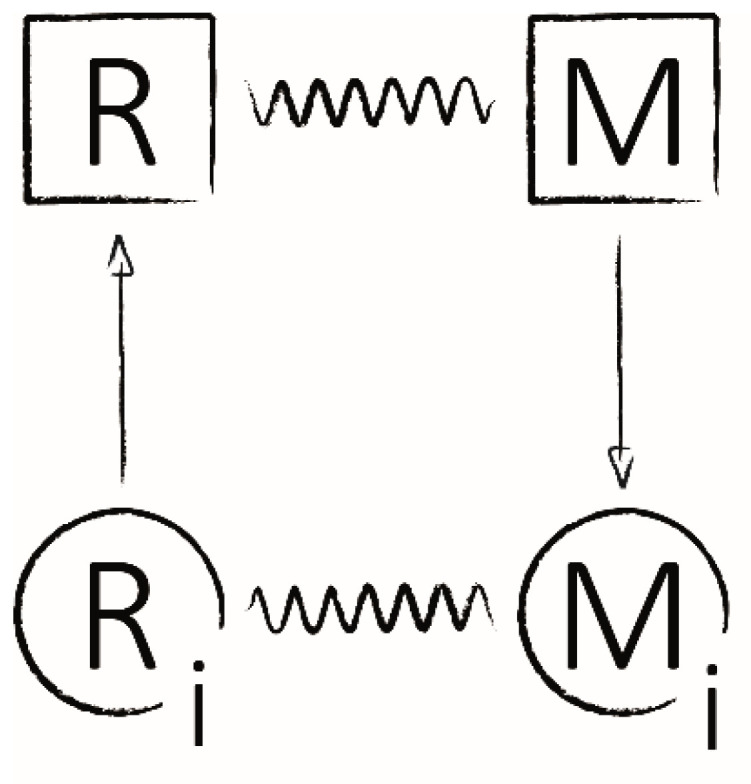
Schematic model of rhodopsin states and their conversions. Simplified cycle of theoretical state conversions of rhodopsin during photoactivation (R to M), deactivation (M to M_i_) and regeneration (M_i_ to R_i_ to R). Sinus waves represent photoconversions (R to M, M to R, M_i_ to R_i_ and R_i_ to M_i_), arrows depict metabolic reactions (M to M_i_ and R_i_ to R). Adapted from [26].

**Figure 3 ijms-23-14674-f003:**
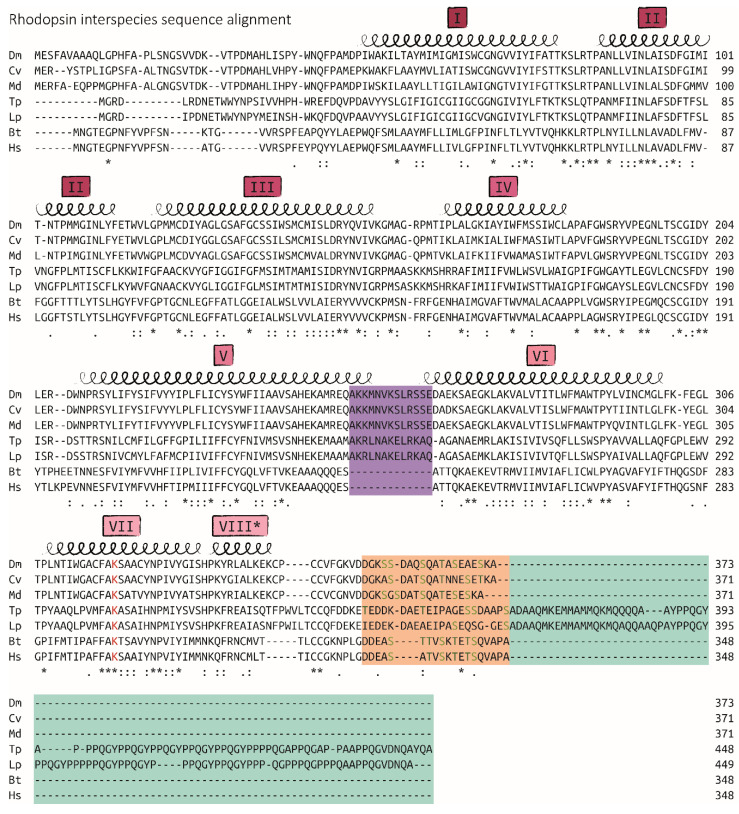
Structure and sequence comparison of rhodopsin from various species of invertebrates and vertebrates. Multiple sequence alignment of rhodopsin from invertebrate (three fly, two squid species) and vertebrate (cattle, human) species mainly discussed in this article and abbreviated as Dm: *Drosophila melanogaster*, Cv: *Calliphora vicina*, Md: *Musca domestica*, Tp: *Todarodes pacificus*, Lp: *Loligo pealeii*, Bt: *Bos taurus*, Hs: *Homo sapiens*. Sequences are denoted in amino acid one letter codes, dashes represent gaps. Alignments were performed and similarities evaluated by EMBL-EBI Clustal Omega [34]. Consensus symbols are shown below the sequences: asterisks (*) indicate fully conserved residues, colons (:) indicate residues with strongly similar properties, periods (.) indicate residues with weakly similar properties. Spirals above the sequences depict regions that fold into α-helices and are numbered sequentially with roman numerals of the seven transmembrane helices (I–VII) of GPCRs. The eighth short helix (VIII *) is not membrane-spanning, but rather laterally membrane-associated. Helices V and VI are slightly longer than the others and extend from the membrane into the cytoplasm. This is more pronounced in invertebrate rhodopsin due to additional amino acid residues shaded with a purple background. The central lysine residue (K) in helix VII which forms a Schiff base with the chromophore retinal is indicated in red. Weakly conserved C-termini are shaded with an orange background and harbor serine (S, light green) and threonine (T, dark green) residues as putative phosphorylation sites. Cephalopod rhodopsins feature significantly longer proline-rich C-termini which are shaded with a cyan background.

**Figure 4 ijms-23-14674-f004:**
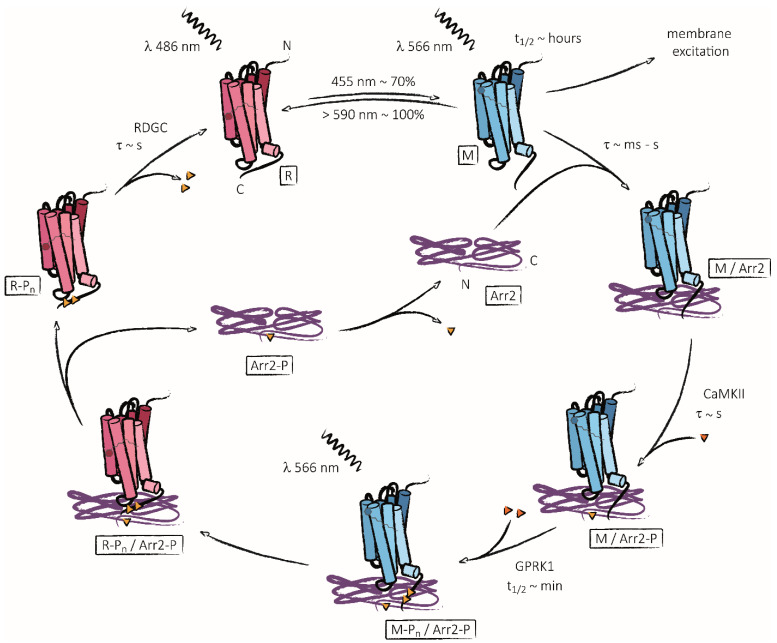
Schematic representation of rhodopsin photoactivation, deactivation and regeneration. *Drosophila* main rhodopsin Rh1 was characterized to possess a light absorption spectrum that peaks around 470–486 nm [68,69,70,71]. Absorption of a photon (depicted as sinus wave) by the chromophore 11-*cis*-3-hydroxyretinal results in the conformational conversion to all-*trans*-3-hydroxyretinal and, thus, from rhodopsin (R, fuchsia) to metarhodospin (M, blue), starting the G protein-coupled phototransduction cascade ending in membrane excitation. The absorption spectrum of M is shifted towards longer wavelengths with its maximum at around 560–580 nm [68,69,70,71]. Photon absorption by metarhodopsin reconverts M back to R. Due to an overlap in their absorption spectra, maximal rhodopsin activation is achieved with a wavelength of 455 nm, converting around 70% of all R molecules towards M [64,65]. Reciprocally, illumination with wavelengths above 590 nm reconvert virtually 100% of M back to R. M has been shown to be thermally stable with a half-life (t_1/2_) of several hours [63,72]. In a physiological context, M is bound in a 1:1 ratio by arrestin 2 (Arr2, purple) with a time constant (τ) in the range of (milli-)seconds which prevents M from further signaling through its G protein [42,51,52,53,54,55,60]. Phosphorylation of M-bound Arr2 is then catalyzed by Ca^2+^-dependent CaMKII in a matter of seconds [56]. With a half-life of a few minutes, Arr2-bound M is subsequently phosphorylated by GPRK1 [25,42]. Upon absorption of another photon, Arr2-bound M-P_n_ is reconverted to R-P_n_, releasing arrestin 2 into the cytoplasm. R-P_n_ is then dephosphorylated, Ca^2+^-dependent, to its original state R by way of RDGC within minutes [25,41,42]. Finally, free Arr2-P is dephosphorylated to Arr2 to close the cycle. ATP is depicted as orange triangles, bound and free phosphate groups as yellow triangles.

**Figure 5 ijms-23-14674-f005:**
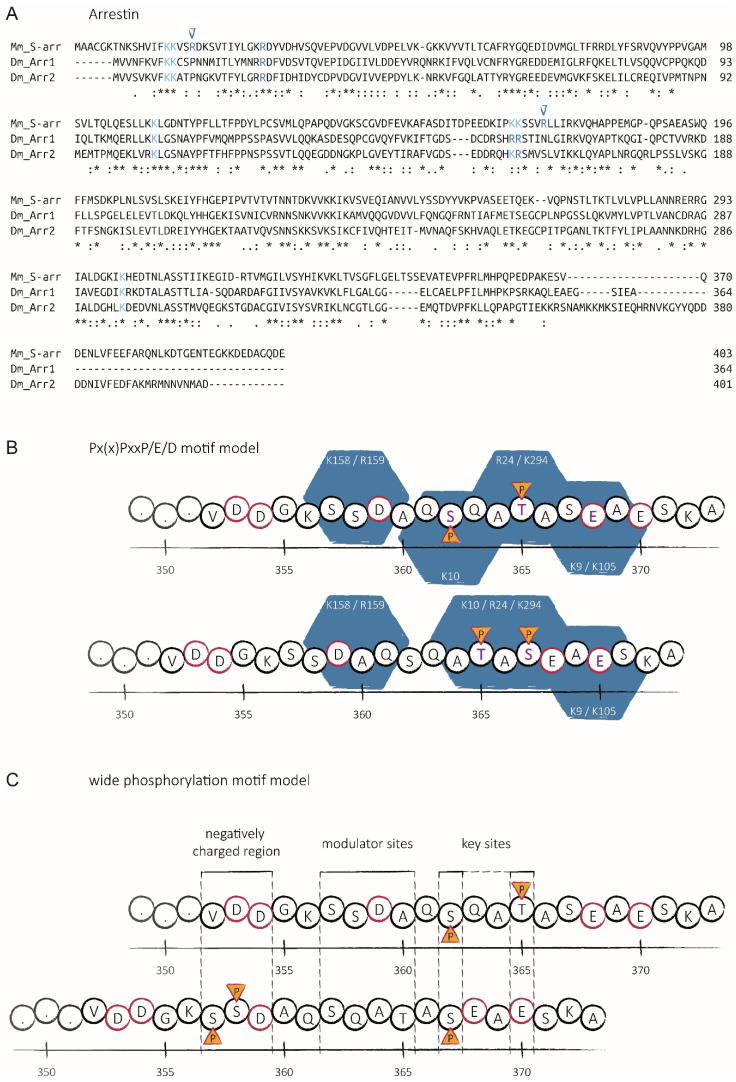
Sequence and motif comparison between vertebrate and invertebrate arrestin–rhodopsin complexes. Sequences are denoted in amino acid one letter codes, dashes represent gaps. Alignments were performed and similarities evaluated by EMBL-EBI Clustal Omega [34]. Consensus symbols are shown below the sequences: asterisks (*) indicate fully conserved residues, colons (:) indicate residues with strong similar properties, periods (.) indicate residues with weak similar properties. (**A**) Sequence alignment of visual arrestin (S-arr) from mouse (*Mus musculus*) with the two arrestins from *Drosophila melanogaster* (Arr1 and Arr2). Lysine (K, cyan) and arginine (R, blue) residues contributing to the positively charged “pockets” are indicated. Important residues (R19 and R172 according to the mural sequence) missing in the fly sequence are marked by blue triangles above. (**B**) Schematic representation of rhodopsin’s C-terminus from *Drosophila melanogaster* with putative phosphorylation sites (P, yellow triangles) to fit the proposed vertebrate barcode motifs. Negatively charged moieties are indicated with dark red contours, positively charged “pockets” as blue hexagons with white labels of contributing amino acid positions (numbering according to Arr2 sequence). Two potential configurations fitting the Px(x)PxxP/E/D motif model (letters of contributing residues shown in purple) [8]. (**C**) Schematic representation of rhodopsin’s C-terminus from *Drosophila melanogaster* with putative phosphorylation sites (P, yellow triangles) to fit the proposed vertebrate barcode motifs. Negatively charged moieties are indicated with dark red contours. Two potential configurations fitting the wide phosphorylation motif model [118]. Putative functional regions in the sequence of *Drosophila* rhodopsin (negatively charged region, modulator sites, key sites) are indicated between vertical dashed lines.

## Data Availability

Not applicable.

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
