# Peer review of "The Role of Reversible Phosphorylation of Drosophila Rhodopsin"

_ijms, 2022, doi:10.3390/ijms232314674_

Round 1

Reviewer 1 Report

In this manuscript, Smylla et al., have provided an insightful review of rhodopsin signaling and signal termination by phosphorylation. They have also provided a comparative analysis of rhodopsin phosphorylation in vertebrate and invertebrate systems. They also explored the role of rhodopsin phosphorylation in arrestin 1 binding. This review will be a good read for researchers in the field of G-protein coupled receptor (GPCR)-arresting signaling. Although, the review is well written, it needs to be modified further before publication. The major concerns have been listed below:

MAJOR CONCERNS:

1.     A section needs to be provided discussing the role of non-canonical arrestin signaling downstream of GPCR phosphorylation and comparing how it is different downstream of rhodopsin-arrestin complex.

2.     The authors need to provide a section summarizing the literature on molecular level changes that are introduced in arrestin structure when they bind to phosphorylated S/T residues in rhodopsin C -tail

Author Response

  1. We agree that non-canonical arrestin signaling is linked to the topic of our review. Therefore, we made an addition at the end of chapter 3.2. summarizing various pathways described in literature. However, since to our knowledge there are no reports about non-canonical arrestin signaling in the context of Drosophila rhodopsin, we did not discuss this any deeper.
  1. We wholly agree that arrestin conformation is a central part of potential downstream signaling. Accordingly, we added an entire chapter 4.1. dedicated to this topic in vertebrates as well as in invertebrates.

We hope that the changes we made satisfy your requests. 

Reviewer 2 Report

This review highlights the effects of rhodopsin phosphorylation and the differences in vertebrate vs. invertebrate rhodopsin functions, espeically when examined in Drosophila.

The review was well written and highlighted many older, some much older, works on the topic.  There was enough recent contributions to this field to warrant this review, however, a lot of the information contained in it, especially for the purpose of background explanaitons, is quite dated.

For critique, the only minor critique I had was that some of the text on the figures was a little too "cartoon-like".  Some of it was a little difficult to read based on the font used.  I would recommend making it a little better to read in the figures, as the figures are very well done and useful.

Author Response

We agree and have changed the font used in the figures to “Calibri” which is more easily legible but still fits the style of the figures.

We hope that this change alleviates your concerns.